# Assessing Large Language Models in Updating Their Forecasts with New Information

## Abstract

Prior work has largely treated future event prediction as a static task, failing to consider how forecasts and the confidence in them should evolve as new evidence emerges. To address this gap, we introduce EVOLVECAST, a framework for evaluating whether large language models appropriately revise their predictions in response to new information. In particular, EVOLVECAST assesses whether LLMs adjust their forecasts when presented with information released after their training cutoff. We use human forecasters as a comparative reference to analyze prediction shifts and confidence calibration under updated contexts. While LLMs demonstrate some responsiveness to new information, their updates are often inconsistent or overly conservative. We further find that neither verbalized nor logits-based confidence estimates consistently outperform the other, and both remain far from the human reference standard. Across settings, models tend to express conservative bias, underscoring the need for more robust approaches to belief updating.

## 1 Introduction

Large language models (LLMs) have achieved strong performance across a wide range of NLP tasks, particularly those involving factual recall and reasoning over known information (Lin et al., 2022; Guo et al., 2023) including temporal information (Ding et al., 2025). However, most evaluations remain static and retrospective: they assess what models know about the world up to their training cutoff. In contrast, real-world decision-making often requires reasoning about uncertain future events, where outcomes are not yet known and relevant information evolves over time.

Forecasting plays a central role in domains such as policymaking, science, and technology. It requires anticipating future developments based on incomplete or uncertain evidence, and adjusting those beliefs as new signals arrive. Unlike fact retrieval tasks, forecasting is inherently temporal and dynamic. For instance, given the question, "Will GPT-5 be released in 2025?", an LLM might answer "Yes" based on existing trends in AI development. However, without a way to incorporate emerging information, such as announcements or delays, such forecasts remain static, and may quickly become outdated or misleading. To illustrate the dynamic nature of forecasting, see the example in Figure 1.

Prior work has begun to explore the forecasting capabilities of LLMs (Jin et al., 2020; Zou et al., 2022; Yuan et al., 2024a). These efforts typically frame forecasting as a static problem, where the model produces a categorical answer based on its existing knowledge. Some studies incorporate external information, including textual evidence such as news (Ye et al., 2024; Halawi et al., 2024), but use this information to improve the model's *final prediction*, not to evaluate how forecasts change over time. More recently, forecasting has been reframed as a probabilistic task, where models are expected to produce confidence scores in addition to predictions (Karger et al., 2024; Yuan et al., 2025). In this setting, the goal is not just to answer the mentioned question with "Yes", but to provide a confidence level, e.g., 70%, that reflects the model's uncertainty. These works though showed that even when models make correct predictions, their confidence is often miscalibrated.

Yet even confidence-aware predictions remain static if models cannot revise their beliefs. In practice, forecasters routinely update their views as new information becomes available. For example, if a forecaster initially assigns a 55% chance to the release of GPT-5 in 2025, but later encounters a credible report stating "Sam Altman has hinted at a major announcement next week," they might revise their estimate to 70%. A capable forecasting model should exhibit similar adaptive behavior, not only producing reasonable confidence scores, but updating them in response to emerging signals.

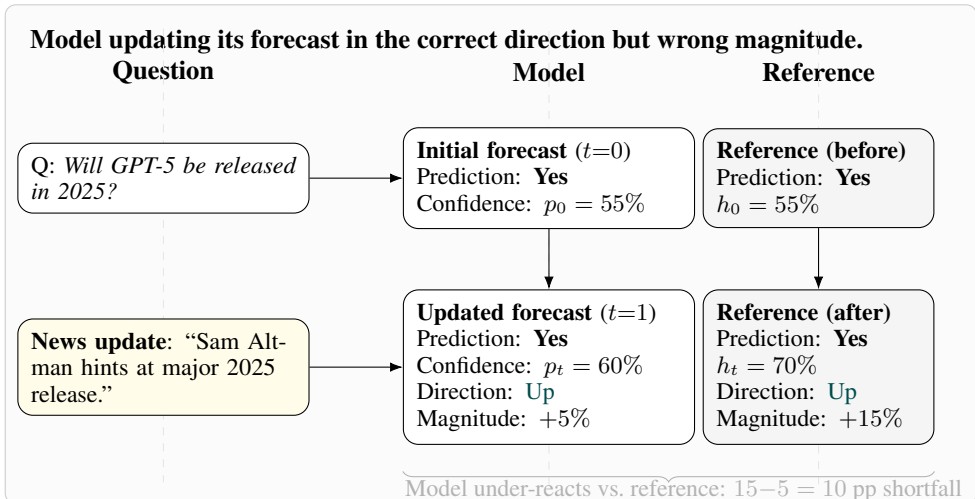

Figure 1: Illustrative example of belief updating in EVOLVECAST. A forecasting question asks: *"Will GPT-5 be released in 2025?"*. Both the model and the human reference start at 55% confidence. After a news update (*"Sam Altman hints at major 2025 release"*), the human reference revises upward to 70% (an increase of 15%), while the model increases only to 60% (an increase of 5%). The model captures the correct *direction* of change but underestimates its *magnitude*, resulting in a 10% shortfall.

Evaluating belief dynamics provides insight into how well models reason under uncertainty and adapt over time, which remains a blind spot in most LLM assessments.

This paper introduces EVOLVECAST, a framework for evaluating whether language models revise their forecasts when presented with new information. Rather than focusing on static predictions, we assess whether models adjust both their predicted outcome and associated confidence in response to emerging evidence. To this end, we construct a benchmark of forecasting scenarios where information relevant to the forecast becomes available after a model's training cutoff. We evaluate LLM predictions with and without access to this new information, and evaluate the resulting belief changes against those made by human forecasters under similar conditions. In an additional setting, we also provide the model with a time series of historical human forecasts up to the moment of the news, allowing it to optionally leverage trends in human belief as a contextual reference without treating it as ground truth. This enables a softer form of conditioning that mimics the information a real-world forecaster might observe, without prescribing correct answers.

Our evaluation focuses on the direction and magnitude of prediction shifts and the calibration of updated confidence scores as exampled in Figure 1. Our findings show that while language models can adjust their forecasts in response to new information, their updates remain limited: belief revisions are often conservative, confidence estimates vary by method without a clear advantage, and both fall short of human reference forecasts. These challenges highlight fundamental difficulties in modeling belief dynamics and underscore the need for evaluation frameworks that extend beyond static forecasting accuracy. EVOLVECAST provides such a framework, offering a principled setting for analyzing how models adapt their predictions as the world evolves.

## 2  RELATED WORK

A growing body of research has investigated the forecasting capabilities of language models, though most efforts emphasize event prediction rather than belief dynamics or confidence calibration. Open-Forecast (Wang et al., 2025) introduces a large-scale benchmark for open-ended multi-step forecasting, but focuses primarily on model accuracy rather than the calibration of confidence estimates. Fore-castBench (Karger et al., 2024) frames forecasting as a probabilistic task with evolving predictions over time, but does not explicitly examine how models adjust their beliefs in light of new information. Time-R1 (Liu et al., 2025) presents a reinforcement learning framework to endow smaller LLMs with temporal understanding and future event generation capabilities, showing strong performance on

forecasting tasks beyond training cutoff. While our focus also involves forecasting under temporal uncertainty, unlike previous we evaluate belief revisions as new signals emerge. Beyond forecasting, several benchmarks target reasoning and plausibility assessment. COPA (Roemmele et al., 2011) and HellaSwag (Zellers et al., 2019) evaluate causal and commonsense reasoning via multiple-choice inference tasks. PRobELM (Yuan et al., 2024b) examines models' ability to rank outcomes by plausibility, drawing on general world knowledge. While these evaluations probe important reasoning capabilities, they do not test how models reason under uncertainty or update beliefs over time.

## 3 EVOLVECAST: DYNAMIC FORECASTING EVALUATION FORMALIZATION

### 3.1 TASK DEFINITION

We formalize EVOLVECAST as a dynamic forecasting task, where the goal is to evaluate whether a language model revises its predictions and confidence in response to new information in a manner comparable to a reference standard. Let $q$ denote a forecasting question about a future event (e.g., "Will GPT-5 be released in 2025?"). At time $t$, new information $x_t$ (e.g., a news headline or update) becomes available, and we measure how the model's belief changes when conditioned on it. For binary questions ($\mathcal{Y} = \{\text{Yes}, \text{No}\}$), the model's belief at time $t$ reduces to a scalar confidence $p_t = P_t(\text{Yes} \mid x_t) \in [0, 1]$, and the belief update is defined as $\Delta p = p_t - p_0$.

### 3.2 EVALUATION CRITERIA

We evaluate the quality of model belief updates by comparing the change in model confidence, $\Delta p = p_t - p_0$, to the corresponding reference shift $\Delta h = h_t - h_0$, on three complementary aspects:

**Directional Agreement.** We use *Mean Directional Accuracy (MDA)* to assess whether the model and reference forecasts tend to update their beliefs in the same direction across examples. For a set of $N$ instances, MDA is defined as:

$$\text{MDA} = \frac{1}{N} \sum_{i=1}^{N} 1 \left[ \text{sign}(\tilde{\Delta}_p^{(i)}) = \text{sign}(\tilde{\Delta}_h^{(i)}) \right],$$

where $\Delta p_i$ and $\Delta h_i$ are the model and reference forecast deltas for the $i$-th instance. We apply a minimal change threshold $\epsilon$ to avoid counting negligible shifts as meaningful updates. Specifically, we treat both $\Delta p_i$ and $\Delta h_i$ as zero when $|\Delta| < \epsilon$, and consider the signs to match in such cases. Let $\tilde{\Delta}_p$ denote the thresholded version of $\Delta p$, where $\tilde{\Delta}_p = 0$ if $|\Delta p| < \epsilon$, and similarly for $\Delta h$. *Precision, Recall, and F1 (micro-averaged)* are also computed binary target by treating "directional match" as the positive class (i.e., TP: signs match; FP: signs match when the reference does not; FN: signs do not match when the reference does). In practice, this means micro-averaged recall is always equal to the MDA, since both reduce to the proportion of correctly classified instances; in addition, this reduces each update to one of three labels: `Up` ($\Delta > \epsilon$), `Down` ($\Delta < -\epsilon$), or `Still` ($|\Delta| < \epsilon$); and we evaluate whether the model's label matches the reference.

**Magnitude Alignment.** We measure how closely the magnitude of the model's belief update matches that of reference forecasters. We report two complementary metrics: *Mean Squared Error (MSE)* and *Symmetric Mean Squared Percentage Error (SMSPE)*. The MSE is defined as:

$$\text{MSE} = \frac{1}{N} \sum_{i=1}^{N} (\Delta p_i - \Delta h_i)^2,$$

To evaluate relative alignment, we define the symmetric percentage change in belief as:

$$\delta_p^{(i)} = \frac{p_t^{(i)} - p_0^{(i)}}{(p_t^{(i)} + p_0^{(i)})/2}, \quad \delta_h^{(i)} = \frac{h_t^{(i)} - h_0^{(i)}}{(h_t^{(i)} + h_0^{(i)})/2},$$

where $p_0^{(i)}$ and $p_t^{(i)}$ are the model's confidence before and after seeing new information (similarly for the reference forecasts), similar to MAPE (De Myttenaere et al., 2016). The SMSPE is given by:

$$\text{SMSPE} = \frac{1}{N} \sum_{i=1}^{N} \left( \delta_p^{(i)} - \delta_h^{(i)} \right)^2.$$

**Confidence Calibration.**    To assess whether models produce well-calibrated confidence estimates, we compute the *Brier Score* (Brier, 1950). This allows us to evaluate whether the model's confidence aligns with reference before and after new information is introduced. The Brier score between model prediction $p$ and reference forecast $h$ is defined as: $\text{Brier}(p, h) = (p - h)^2$. We report the change in calibration directly as the average difference in squared errors before and after conditioning on new information:

$$\Delta \text{Brier} = \frac{1}{N} \sum_{i=1}^{N} \left[ (p_t^{(i)} - h_t^{(i)})^2 - (p_0^{(i)} - h_0^{(i)})^2 \right].$$

A negative $\Delta \text{Brier}$ indicates improved calibration after observing new information (since lower Brier is better), while a positive value suggests degradation.

## 4    EVOLVECAST DATASET CONSTRUCTION

EVOLVECAST is constructed from Metaculus (`www.metaculus.com`; see examples in Appendix C), an online forecasting platform where users submit probability estimates to questions spanning domains such as politics, economics, health, and technology. Metaculus aggregates these into a community prediction that is continuously updated until shortly before resolution. Each question is associated with predefined resolution criteria, and the platform enforces strict guidelines to ensure that forecasts are consistent with them and human-generated and consistent with those criteria. Metaculus also has a strong empirical track record: over the past five years, community predictions on resolved binary questions have shown close alignment to observed outcomes, with roughly half of observed frequencies lying within the 90% credible interval around the ideal calibration line.[1] This calibration performance provides confidence that the platform's aggregated predictions are reliable.

We apply several filtering steps to ensure data quality. First, we discard questions that Metaculus marks as ambiguous or that are resolved in ways that make forecasting infeasible. For example, when the outcome falls outside the prediction range or when resolution criteria are modified after submission. Second, we only include questions with at least 100 individual forecasts to maintain statistical reliability in the aggregated reference prediction. Finally, we restrict the dataset to binary (Yes/No) questions. The resulting dataset consists of 1,613 question–news pairs with timestamp.

### 4.1    REFERENCE FORECASTS AND CONFIDENCE EXTRACTION

In our setting, ground-truth confidence values are not directly observable. We therefore follow prior work that uses human forecast distributions as a proxy for uncertainty (Plank, 2022; Baan et al., 2024; Yuan et al., 2025). The intuition is that aggregated human forecasts, even when forecasters disagree, provide informative signals of uncertainty: disagreement itself reflects the inherent ambiguity and difficulty of the task, similar to how disagreement among annotators is leveraged in natural language tasks with multiple plausible interpretations. This approach has several advantages. First, it captures the fact that some forecasting questions are inherently uncertain, so variance across forecasters represents a genuine property of the task rather than annotation noise. Second, aggregation over many individuals smooths out idiosyncratic biases while preserving collective uncertainty, yielding a stable yet informative signal. Third, these distributions are updated dynamically as new evidence appears, making them particularly well suited for evaluating belief revision.

For Boolean questions, where the community assigns probability $P$ to the correct outcome, we compute a normalized confidence score as $h = \sigma\left(\frac{\ln P - \ln 0.5}{\ln 2}\right)$. This transformation ensures that confidence is scaled relative to a chance-level (50%) baseline. While aggregated forecasts are sometimes incorrect, they nonetheless represent the best available reasoning given the information at the time, and thus serve as a practical and informative reference point for evaluating model confidence.

### 4.2    NEWS RETRIEVAL AND ALIGNMENT

In addition to questions and forecasts, EVOLVECAST pairs each instance with a contemporaneous news update. The key challenge is to determine when new information becomes available that could plausibly shift forecasts. We detect these moments by monitoring the comment streams associated

---

[1] `https://www.metaculus.com/questions/track-record`

with each Metaculus question: whenever a new comment is posted, we treat this as a candidate signal that new evidence has emerged. To reduce noise, we filter out bot-generated or automated comments as well as very short entries (fewer than 20 characters). Once a candidate timestamp is identified, we search for relevant news articles within a one-week window preceding the comment to account for possible delays between publication of external information and the time at which it is reflected in forecaster discussion. News articles are retrieved via the Google Search API, using the original question as the query, and for each candidate timestamp we collect up to 100 related articles.

To select the most relevant update, we compute semantic similarity between the comment text and each retrieved article using sentence embeddings (Reimers & Gurevych, 2019). Articles are then ranked by similarity score, and by default we retain the top-ranked article, storing both its title and headline as the associated news update. This procedure ensures that each question–forecast pair is aligned with a news item that is both temporally plausible and semantically linked to the corresponding forecaster discussion. Because a single question may be associated with multiple updates over time, the dataset may contain repeated questions paired with different news events. In total, this process yields 1,613 aligned question–news pairs across domains.

# 5 EXPERIMENTAL SETUP

## 5.1 MODELS

We evaluate three openly available *reasoning* models from the DeepSeek-R1 series (Guo et al., 2025): `DeepSeek-R1-Distill-Qwen-1.5B`, `DeepSeek-R1-Distill-Qwen-7B`, and `DeepSeek-R1-Distill-LLaMA-8B`, together with their published *base* counterparts from the same backbones (Qwen 2.5 (Bai et al., 2025) and Llama 3.1 (Dubey et al., 2024) families, as indicated in the official model cards).[2] This choice serves two purposes. First, it provides coverage across both Qwen and Llama backbones and multiple parameter scales (1.5B/7B/8B), enabling us to probe how *scale* and *architecture* relate to belief-updating behavior. Second and critically, it allows us to isolate the effect of *reasoning-style post-training* (e.g., distillation/RL procedures specific to R1) by comparing each R1 model against a matched base model. Including unrelated weaker models would introduce confounds (different tokenizers, pretraining corpora, and objectives) that obscure the contribution of reasoning post-training and inflate variance without strengthening causal claims about update dynamics. All models are evaluated with identical prompts and decoding settings.

## 5.2 INFERENCE PROCEDURE

For each question, we run two conditions with identical instructions: (i) a *question-only* condition (no external update) and (ii) a *question+news* condition where we augment the prompt with a retrieved news snippet aligned to the question (Sec. 4). Full details are provided in Appendix D.

Because the exact training cutoffs of these models are undisclosed, we restrict evaluation to questions first posted strictly *after October 2023*. For each instance we use two explicit evaluation dates: $T_0$ is the question date (the prompt states "today is $T_0$"), and $T_1$ is the publication date of the associated news update (the prompt states "today is $T_1$" and includes the dated snippet), with $T_0 < T_1$. In both conditions the prompt instructs the model to use only information available up to the stated date (e.g., "You do not have access to updates after $T$").

This dynamic anchoring serves two goals. First, it closely mirrors the real forecaster workflow: form a view at $T_0$, then revise at $T_1$ when new evidence arrives, thereby avoiding hindsight effects. Second, it minimizes leakage concerns and unknown-cutoff confounds: our primary metrics evaluate the *within-instance change* $\Delta p = p_t - p_0$ from $T_0$ to $T_1$. By comparing the same model on the same question before vs. after dated evidence, we largely cancel effects of any static knowledge in pretraining and directly test responsiveness to new information rather than recall of facts.

---

[2]We pair each R1-distilled model with the corresponding base instruction model from its backbone family following the mapping described in the providers' model cards.

### 5.3 CONFIDENCE EXTRACTION

Following prior work on eliciting model self-assessments (Xiong et al., 2023), we evaluate model confidence using two complementary approaches: its own uncertainty and underlying probabilities.

**Black-box (verbalized) confidence.** The model is instructed to provide a binary prediction ("Yes" or "No") and to assign a probability on a 1–10 scale with descriptive anchors ("1: extremely unlikely" ... "10: extremely likely"). We normalize this verbalized score to $[0, 1]$ for evaluation. This setting aligns with how human forecasters typically express uncertainty through explicit probability estimates, facilitating direct comparison with the aggregated human reference.

**White-box (logit-based) confidence.** We also derive confidence estimates directly from the model's output probabilities. For a single generated answer $y = (y_1, \ldots, y_T)$, we compute the mean token probability $\hat{p} = \frac{1}{T} \sum_{t=1}^{T} P(y_t \mid y_{<t}, x)$, where $P(y_t \mid y_{<t}, x)$ is the model's predicted probability of token $y_t$ given the context. This logit-based measure reflects the model's internal certainty about its produced sequence and provides a white-box perspective complementary to the verbalized estimates.

## 6 RESULTS

### 6.1 BLACK- VS. WHITE-BOX CONFIDENCE: NO CLEAR WINNER, BUT REASONING HELPS

Table 1 compares verbalized (black-box) and logit-based (white-box) confidence across reasoning-tuned and base models. Overall, neither approach emerges as a clear winner: both exhibit low directional agreement and poor calibration relative to human forecasters. **Reasoning-tuned models consistently outperform their base counterparts, particularly at larger scales (7B and 8B), yet within each backbone the gap between black- and white-box methods remains small and inconsistent.** Several factors likely contribute to the underperformance of both approaches. Verbalized probabilities show slightly greater stability across settings, yet they remain imperfect: models often default to conservative or generic values, which blunts the fidelity of their probability estimates. Logit-based confidences, by contrast, draw more directly on internal activations but are highly sensitive to prompt length, decoding hyperparameters, and normalization schemes, which can undermine robustness. Neither approach therefore provides a consistently reliable signal, and small numerical differences between them should be interpreted with caution.

Beyond the mechanics of confidence elicitation, forecasting itself poses distinctive challenges for LLMs. The task demands not only access to broad domain knowledge but also the capacity to integrate novel information and update beliefs in a calibrated fashion. Current models show limited ability to perform these updates reliably, leading to systematic miscalibration even when directional reasoning is sound. This pattern echoes prior findings (Karger et al., 2024), which likewise identified forecasting as a setting that exposes fundamental weaknesses in both reasoning and calibration. To better understand these dynamics, we narrow our subsequent ablations to a subset of settings that more clearly isolate model behavior under controlled conditions. Even with this refinement, the central result remains unchanged: **LLMs are far from human-like in belief updating, regardless of whether confidence is elicited in black-box or white-box form.**

### 6.2 ABLATION: ACCUMULATED NEWS CONTEXT

Table 2 reports results when models are given access to the *entire sequence of news updates from $T_0$ to $T_1$* (Accumulated) rather than only the latest update (Single). This setting tests whether richer temporal context improves belief updating. A single forecasting question can be associated with multiple news items over time, and the accumulated condition passes all updates in chronological order, while the single condition isolates only the most recent update.

To focus the analysis and other ablation studies, we report results using verbalized confidence only, since in the main setting it showed broadly similar patterns to logit-based confidence but with slightly more stable behavior. We also restrict attention to the DeepSeek R1 reasoning models, as the ablation targets temporal context rather than differences between reasoning and base backbones.

Table 1: Main results. Top: **black-box** (verbalized) confidence. Bottom: **white-box** (logit-based) confidence. Columns are grouped as *Directional agreement* (MDA/Prec/Rec/F1), *Magnitude error* (MSE/SMSPE), and *Calibration change* ($\Delta$Brier; negative is better).

Black-box (verbalized) confidence

| Model | Directional agreement | | | | Magnitude | | Cal. |
|---|---|---|---|---|---|---|---|
| | MDA | Prec | Rec | F1 | MSE | SMSPE | $\Delta$Brier |
| *Qwen backbone* | | | | | | | |
| DS R1 Qwen-1.5B | 0.2529 | 0.4413 | 0.2529 | 0.2559 | 0.0258 | 0.1033 | 0.0232 |
| Qwen-2.5 1.5B | 0.2372 | 0.4768 | 0.2372 | 0.2455 | 0.0262 | 0.1059 | 0.0238 |
| DS R1 Qwen-7B | 0.3534 | 0.4639 | 0.3534 | 0.3791 | 0.0232 | 0.0939 | 0.0210 |
| Qwen-2.5 7B | 0.2603 | 0.4361 | 0.2603 | 0.2654 | 0.0256 | 0.1015 | 0.0231 |
| *LLaMA backbone* | | | | | | | |
| DS R1 LLaMA-8B | 0.3360 | 0.4710 | 0.3360 | 0.3607 | 0.0237 | 0.0947 | 0.0214 |
| LLaMA-3.1 8B | 0.2199 | 0.3003 | 0.2199 | 0.2063 | 0.0267 | 0.1067 | 0.0244 |

White-box (logit-based) confidence

| Model | Directional agreement | | | | Magnitude | | Cal. |
|---|---|---|---|---|---|---|---|
| | MDA | Prec | Rec | F1 | MSE | SMSPE | $\Delta$Brier |
| *Qwen backbone* | | | | | | | |
| DS R1 Qwen-1.5B | 0.2461 | 0.4740 | 0.2461 | 0.2325 | 0.0260 | 0.1050 | 0.0236 |
| Qwen-2.5 1.5B | 0.2298 | 0.4505 | 0.2298 | 0.2360 | 0.0266 | 0.1065 | 0.0241 |
| DS R1 Qwen-7B | 0.3440 | 0.4422 | 0.3440 | 0.3650 | 0.0237 | 0.0951 | 0.0217 |
| Qwen-2.5 7B | 0.2545 | 0.4228 | 0.2545 | 0.2592 | 0.0259 | 0.1022 | 0.0234 |
| *LLaMA backbone* | | | | | | | |
| DS R1 LLaMA-8B | 0.3271 | 0.4595 | 0.3271 | 0.3510 | 0.0242 | 0.0960 | 0.0219 |
| LLaMA-3.1 8B | 0.2144 | 0.2925 | 0.2144 | 0.2018 | 0.0273 | 0.1082 | 0.0249 |

Interestingly, the results in Table 2 show no consistent improvement from providing accumulated news; in several cases, directional agreement metrics even decline relative to the single-update condition. While the smallest model (Qwen-1.5B) shows minor gains in some metrics under the accumulated setting, these do not generalize: both Qwen-7B and LLaMA-8B degrade when given the full sequence of updates. One possible explanation is that language models struggle to weigh multiple temporally ordered snippets, often overemphasizing earlier or less relevant updates rather than the most recent signal. Unlike human forecasters, who can prioritize and discount information dynamically, models may treat all updates with similar salience, leading to diluted or inconsistent belief adjustments. Another factor is prompt length: longer contexts may increase the chance of distraction or generic reasoning, reducing the precision of updates. Together, these results suggest that **simply concatenating news updates does not guarantee better adaptation, and highlights the challenge of teaching models to integrate evolving evidence effectively.**

The confusion matrices in Fig. 2 provide a complementary perspective. Under the Single condition, all three models already show systematic confusion between "Up" and "Down," with relatively few accurate "Still" predictions. In the Accumulated condition, this imbalance becomes more pronounced: across Qwen-7B and LLaMA-8B, the proportion of "Still" cases misclassified as movement (either "Up" or "Down") increases noticeably. This suggests that when exposed to multiple temporally ordered snippets, models develop a bias toward interpreting evidence as directional even when the reference forecasters remained stable. In other words, **rather than integrating signals over time, the models appear to accumulate noise, amplifying small fluctuations into spurious updates.** The heatmaps therefore reinforce the quantitative results in Table 2: accumulated news does not improve consistency with the human reference and in some cases drives models further away from

Table 2: Ablation on **news context**. Models are evaluated with either only the most recent update (S = Single) or the full sequence of updates from $T_0$ to $T_1$ (A = Accumulated). Metrics are grouped as in Table 1. Results for magnitude error and calibration change will be added once available.

| Model | Directional agreement | | | | Magnitude error | | Calibration |
|---|---|---|---|---|---|---|---|
| | MDA | Prec | Rec | F1 | MSE | SMSPE | $\Delta$Brier |
| DS R1 Qwen-1.5B (S) | 0.2529 | 0.4413 | 0.2529 | 0.2559 | 0.0258 | 0.1033 | 0.0232 |
| DS R1 Qwen-1.5B (A) | 0.2554 | 0.4564 | 0.2554 | 0.2512 | 0.0262 | 0.1052 | 0.0237 |
| DS R1 Qwen-7B (S) | 0.3534 | 0.4639 | 0.3534 | 0.3791 | 0.0232 | 0.0939 | 0.0210 |
| DS R1 Qwen-7B (A) | 0.3236 | 0.4565 | 0.3236 | 0.3473 | 0.0251 | 0.1024 | 0.0228 |
| DS R1 LLaMA-8B (S) | 0.3360 | 0.4710 | 0.3360 | 0.3607 | 0.0242 | 0.0960 | 0.0219 |
| DS R1 LLaMA-8B (A) | 0.3013 | 0.4687 | 0.3013 | 0.3196 | 0.0268 | 0.1083 | 0.0247 |

Figure 2: Normalized confusion matrices for DeepSeek R1 models under Single (S) and Accumulated (A) news updates. Columns correspond to models; rows correspond to evidence settings. Values are column-normalized, showing $\Pr(\text{pred} \mid \text{true})$ in %.

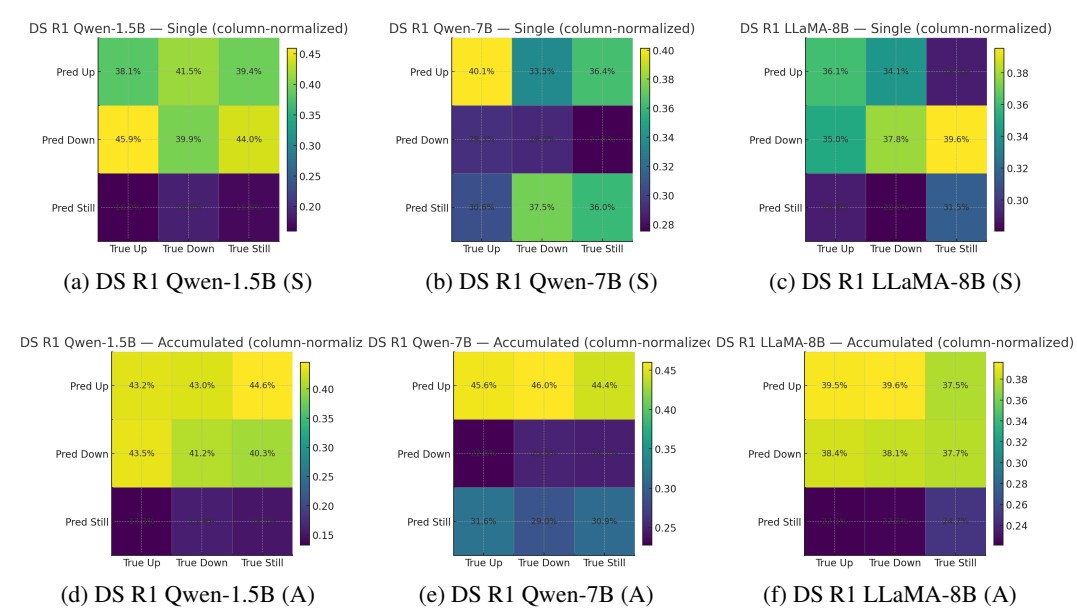

(a) DS R1 Qwen-1.5B (S)  (b) DS R1 Qwen-7B (S)  (c) DS R1 LLaMA-8B (S)

(d) DS R1 Qwen-1.5B (A)  (e) DS R1 Qwen-7B (A)  (f) DS R1 LLaMA-8B (A)

calibrated, conservative updating. See Appendix E.1 for *delta* heatmaps (A−S) that visualize how confusion mass shifts when moving from Single to Accumulated context.

## 6.3 ABLATION: DIRECTIONAL QA PROMPTING

In this ablation, instead of eliciting probabilities before and after a news update and then computing their difference, we directly prompt the model to predict whether the news should make the forecast go "Up," "Down," or remain "Still." This reduces the task to a single classification run rather than two probability-estimation runs, and removes dependence on the quality of numeric calibration. Prompt templates are provided in Appendix F. The motivation for this setting is twofold. First, subtraction of two noisy probability estimates is fragile: even when the direction of change is clear, imperfect calibration or over/under-confidence can obscure it. Second, this format mirrors how human forecasters often reason qualitatively, e.g., "this headline makes outcome $X$ more likely," without attaching precise numbers. Direct directional prompting therefore probes whether LLMs can at least capture the *sign* of belief updates, independent of their ability to express calibrated probabilities.

Table 3: Directional QA format results under the *Single* (S) and *Accumulated Updates* (A) settings. Metrics cover only directional agreement since this setting directly elicits Up/Down/Still labels.

| Model | MDA | Prec | Rec | F1 |
|---|---|---|---|---|
| DS R1 Qwen-1.5B (S) | 0.3521 | 0.4401 | 0.3521 | 0.3762 |
| DS R1 Qwen-1.5B (A) | 0.3620 | 0.4366 | 0.3620 | 0.3838 |
| DS R1 Qwen-7B (S) | 0.4675 | 0.4538 | 0.4675 | 0.4581 |
| DS R1 Qwen-7B (A) | 0.4314 | 0.4387 | 0.4314 | 0.4320 |
| DS R1 LLaMA-8B (S) | 0.4923 | 0.4621 | 0.4923 | 0.4714 |
| DS R1 LLaMA-8B (A) | 0.4389 | 0.4383 | 0.4389 | 0.4351 |

Table 3 reports directional agreement metrics under this setting. Compared to the probability-based approach, the direct method yields markedly higher MDA and F1, especially for larger models. For example, DS R1 LLaMA-8B achieves nearly 0.49 MDA with single updates, compared to only 0.34 under verbalized confidence (cf. Table 1). This indicates that models are substantially better at reasoning about the qualitative *direction* of change than at producing well-calibrated probabilities. Similar as before, accumulated news again does not improve performance: both RS R1 Qwen-7B and RS R1 LLaMA-8B see declines in MDA and F1 when given the full sequence of updates. Together with Sec. 6.2, this highlights that concatenating news does not help models integrate evidence and may in fact dilute the signal. **Overall, this ablation suggests that while LLMs struggle with precise probability calibration, they are relatively more reliable at qualitative directional reasoning.**

To further analyze model behavior, we inspect the confusion matrices of directional predictions under both Single and Accumulated settings (Appendix E.2, Fig. 6). The patterns reveal distinct biases across scales. In the Single setting, DS R1 Qwen-7B and DS R1 LLaMA-8B strongly favor the "Still" label, indicating a conservative updating tendency, while DS R1 Qwen-1.5B behaves more noisily, frequently predicting "Up" even when the true direction is "Down" or "Still." When moving to the Accumulated condition, the conservative bias in the larger models persists but is accompanied by an increased tendency to predict "Up" (e.g., for DS R1 Qwen-7B, predictions of "Up" rise from 221 to 278 on "True Still" cases). For DS R1 Qwen-1.5B, the confusion remains scattered, though its excessive "Up" predictions on "True Still" instances decrease slightly. Overall, this suggests a systematic pattern: smaller models tend to be noisy and overreact, while larger models are conservative but can be nudged into spurious upward shifts when exposed to multiple news updates. This aligns with the quantitative findings that accumulated context does not improve directional agreement and can even reduce it by introducing spurious movement.

## 6.4 ADDITIONAL ABLATIONS

We briefly summarize two further ablations, with full results and visualizations in Appendix G: 1. *Similarity-Sensitive Confidence* implements a semantics-aware estimate (Kuhn et al., 2023) by clustering multiple generations into groups of similar answers and aggregating their confidence. In practice, since our task is binary, the clustering almost always reduces to two groups, limiting its usefulness. While such methods may prove more valuable in open-ended generation settings, here they do not offer additional insight over simpler probability estimates; 2. *Human Forecast Reference as Context* augments prompts with an aggregate human forecast at the corresponding time, providing the model with an explicit calibration anchor. However, we observe no measurable improvements, suggesting that models are unable to effectively exploit even strong external reference signals.

## 7 CONCLUSION

We introduced EVOLVECAST, the first framework to assess how language models revise forecasts when new evidence emerges. Across multiple models and extensive ablation studies, we find that updates are often conservative or inconsistent, with neither verbalized nor logit-based confidence clearly superior and both far from human references. These results underscore the challenge of belief updating in current LLMs and the need for more robust approaches to handling evolving evidence.

ETHICS STATEMENT

This work evaluates how large language models update their forecasts when exposed to new information. Our experiments are conducted entirely on publicly available models and datasets. The forecasting questions and human reference data are sourced from Metaculus, an open platform with strict guidelines for question resolution and user conduct. The news snippets paired with questions are retrieved from publicly accessible web sources, and only short excerpts (title and headline) are included for research purposes. No private or sensitive data are used.

We recognize that forecasting research can have downstream societal implications. Forecasts produced by models may influence decision making in domains such as politics, economics, or health. Our goal is not to deploy automated forecasters, but to analyze their current limitations in belief updating and calibration. By highlighting where models fall short relative to human references, we aim to support the responsible use of AI in forecasting contexts and to discourage premature deployment of uncalibrated systems. All results should therefore be interpreted as a diagnostic study of model behavior, not as actionable forecasts.

REPRODUCIBILITY STATEMENT

We have taken several steps to make our study as reproducible as possible. We provide a number of implementation details in the paper, including data construction steps, hyperparameter choices, and prompt templates. We also release the full set of processed question–news pairs used in our experiments, together with cleaned code, as supplementary material.

One limitation is that our news retrieval step relies on the Google Search API. Because search results may vary over time and are influenced by ranking algorithms, exact replication of the retrieval stage cannot be guaranteed. Despite this, we make available the aligned data used in our experiments so that downstream evaluation can be reproduced. We believe these materials, combined with the methodological details provided in the text, give future researchers the necessary resources to replicate and extend our work.

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

## A  LIMITATIONS

Our study is limited in scope in several ways. We focus on binary forecasting questions, which provides clarity for evaluation but does not cover all types of forecasting tasks. We also evaluate a small set of openly available reasoning models, so results should not be overgeneralized. Finally, because the data construction relies on public APIs and human discussion timestamps, exact replication of forecaster information exposure is not possible. These constraints are inherent to working with real-world forecasting data, and we have aimed to minimize their impact in this paper.

## B  THE USE OF LARGE LANGUAGE MODELS

Large language models were used in the preparation of this paper as writing assistants. Specifically, they were employed to refine phrasing, improve clarity, and suggest alternative structures for sections and subsections. Models were also used to draft prompt templates in an iterative process, which were then carefully reviewed, tested, and adjusted by the authors.

## C  EXAMPLE METACULUS QUESTIONS

To illustrate how human forecasts evolve over time, consider a question:

*Q1: Will the US Senate weaken or eliminate the filibuster before January 3, 2029?*

For Q1, Figure 3 shows how community forecasts changed over time, while Figure 4 presents the histogram of the final forecast distribution.

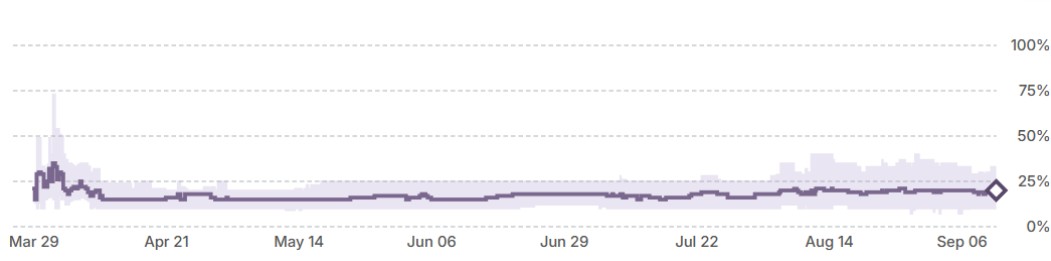

Figure 3: Community prediction trend for a Metaculus question on the US Senate filibuster issue.

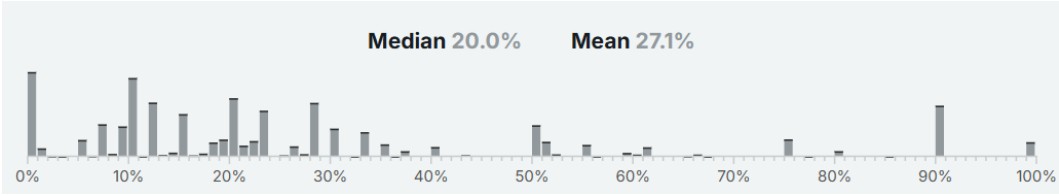

Figure 4: Histogram of final community forecasts.

## D  EXPERIMENTAL DETAILS

### D.1  IMPLEMENTATION DETAILS

**Decoding parameters.** All models are evaluated with identical sampling hyperparameters: `temperature` = 0.6 and `top_p` = 0.95. We set the maximum output length to `max_tokens` = 1024 for verbalized (black-box) confidence runs, and `max_tokens` = 2048 for logit-based (white-box) runs.

**Evaluation thresholds.** Directional agreement metrics (Sec. 3.2) use a minimal change threshold $\epsilon$. For verbalized confidence we set $\epsilon=0$, treating even small probability movements as valid shifts. For logit-based confidence we set $\epsilon=0.05$ to reduce noise from minor fluctuations in token-level probabilities.

**Implementation.** All experiments are run with identical random seeds to ensure consistency across model conditions. In cases where models fail to produce valid JSON outputs, we assign the lowest possible score for that instance. A detailed list of hyperparameters and scripts used for inference are included in the supplementary code release.

## D.2 PROMPT TEMPLATES

We used standardized prompt templates across all models to ensure comparability. Each template contains three blocks: a `<question>` field with the forecasting question, a `<metadata>` field with the evaluation date and optional news update, and a `<task>` field describing the required output format. Prompts also include a `<think>` tag to encourage reasoning before producing the final answer when necessary.

**Verbalized confidence.** The model is instructed to provide a probability estimate on a 1–10 scale with descriptive anchors. We normalize the returned score to $[0, 1]$ for evaluation. An example template is:

```
<question>
${title}
</question>

<metadata>
- Today is ${start_time}.
- You do not have access to updates after ${start_time}.
- A recent news update:  "${best_news}"
</metadata>

<task>
You are an AI model predicting the likelihood of future
events, now incorporating recent news.
Your task is to estimate the probability of the following
event occurring, given the recent news update.

Provide your confidence score on a scale of 1 to 10, where:
1 - Extremely unlikely
2 - Very unlikely
3 - Unlikely
4 - Somewhat unlikely
5 - Neutral (50-50 chance)
6 - Somewhat likely
7 - Likely
8 - Very likely
9 - Extremely likely
10 - Almost certain

Return the confidence score in this format after thinking:
{ "confidence":  X }
</task>
<think>
```

**Logit-based confidence.** The model is prompted to output a binary answer ("Yes"/"No"). Token-level probabilities are then extracted directly from the output distribution. An example template is:

```
<question>
${title}
```

Figure 5: Delta confusion heatmaps (A−S) for **DS R1** models. Each plot shows the difference between column-normalized confusion matrices under Accumulated vs. Single updates, i.e., changes in $\Pr(\text{pred} \mid \text{true})$ (percentage points). Positive values indicate increased mass under Accumulated; negative values indicate decreased mass.

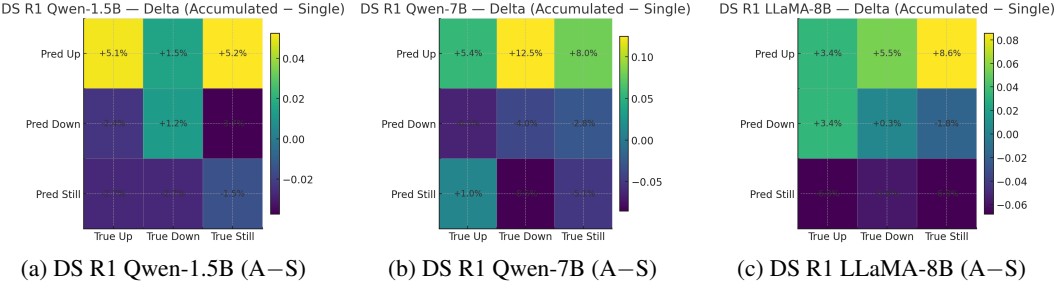

(a) DS R1 Qwen-1.5B (A−S)      (b) DS R1 Qwen-7B (A−S)      (c) DS R1 LLaMA-8B (A−S)

```
</question>

<metadata>
- Today is ${start_time}.
- You do not have access to updates after ${start_time}.
- A recent news update:  "${best_news}"
</metadata>

<task>
You are an AI model predicting the likelihood of future
events, now incorporating recent news.
Your task is to answer if the following event will occur,
given the recent news update.
You must also provide an answer with your best guess.

Return the answer in this format after thinking:
{ "answer":  "Yes" / "No" }
</task>
<think>
```

# E ADDITIONAL VISUALIZATIONS

## E.1 DELTA HEATMAPS: ACCUMULATED MINUS SINGLE

Figure 5 visualizes the difference between Accumulated and Single news contexts as *delta* heatmaps (A−S). Blue cells indicate reduced probability mass in the accumulated condition, while red cells indicate increased mass. Across models, the largest shifts occur in the "Still" column: accumulated updates tend to reduce correct "Still" predictions and redistribute probability into "Up" or "Down." This pattern complements the quantitative results in Table 2 and the confusion matrices in Fig. 2, confirming that accumulated context often introduces spurious directional movement rather than improving alignment with the reference.

## E.2 ADDITIONAL VISUALIZATIONS FOR DIRECT DIRECTIONAL PROMPTING

The visualization complements Table 3 by highlighting systematic biases in how models allocate their predictions. The smaller DS R1 1.5B model produces noisy outputs and often overpredicts "Up," even when the true label is "Down" or "Still." By contrast, DS R1 7B and 8B show a strong conservative bias toward "Still" in the Single setting, but under Accumulated context they shift toward predicting "Up" more frequently. Together, these trends illustrate how accumulated evidence can introduce spurious movements and reduce directional accuracy, especially in larger models.

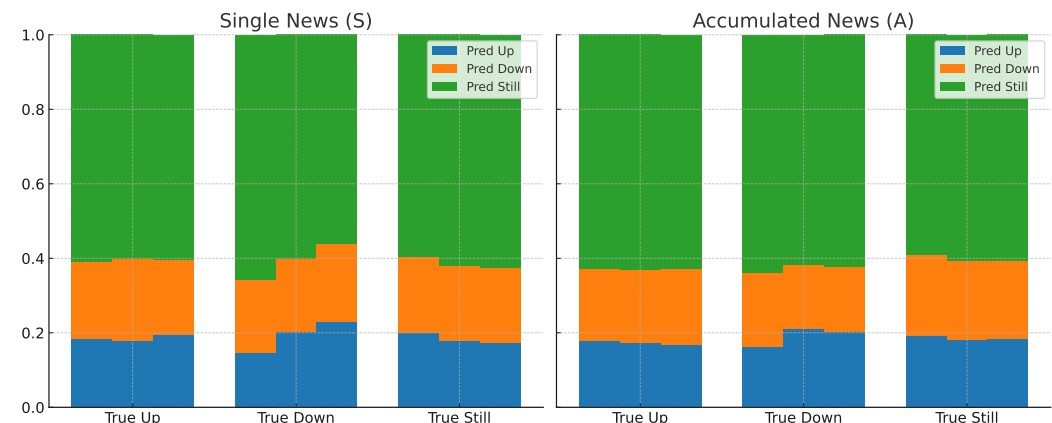

Figure 6: Normalized predicted distributions for the **Directional QA** setting. Bars show the distribution of predicted labels (Up = blue, Down = orange, Still = green) for each true label across DS R1 models under Single (S) and Accumulated (A) news conditions.

## F  DIRECT DIRECTIONAL PROMPTING TEMPLATE

For the direct directional prompting setting, models are asked to classify the impact of a news update as "Up," "Down," or "Still." This approach removes the need to compare before/after probabilities and directly probes whether the model can identify the directional effect of new information. The exact prompt is shown below:

```
<question>
${title}
</question>

<metadata>
- Today is ${start_time}.
- You do not have access to updates after ${start_time}.
- A recent news update:  "${best_news}"
</metadata>

<task>
You are an AI model analyzing how recent news impacts event
predictions.
Your task is to determine whether the confidence in this
event occurring should increase, decrease, or remain the
same after seeing the news.

Return the predicted_trend in this format after thinking:

{ "trend":  "Up" / "Down" / "Still" }
</task>
<think>
```

This minimal QA-style interface makes the task closer to classification benchmarks and avoids dependence on numeric probability estimation, which often suffers from poor calibration.

## G  ADDITIONAL ABLATIONS

### G.1  SIMILARITY-SENSITIVE CONFIDENCE

In this ablation, we implement a semantics-aware confidence estimation method inspired by clustering-based approaches (Kuhn et al., 2023). The intuition is that if a model generates multiple plausible

Table 4: Comparison of logits-based confidence with and without semantic clustering for DS R1 Qwen-1.5B. Metrics are directional agreement (MDA/Prec/Rec/F1).

| Method | MDA | Prec | Rec | F1 |
|---|---|---|---|---|
| Logits (S) | 0.252 | 0.494 | 0.252 | 0.236 |
| Logits (A) | 0.259 | 0.491 | 0.259 | 0.236 |
| Logits+Clust. (S) | 0.247 | 0.442 | 0.247 | 0.239 |
| Logits+Clust. (A) | 0.259 | 0.471 | 0.259 | 0.260 |

answers, the distribution of these generations can provide a more robust uncertainty estimate than any single output. Concretely, we sample $N$ candidate outputs $\{y^{(1)}, \ldots, y^{(N)}\}$ and group them into $K$ clusters based on cosine similarity of sentence embeddings (Reimers & Gurevych, 2019). Confidence for cluster $C_k$ is then defined as

$$\hat{p}_k = \frac{\sum_{i:y^{(i)} \in C_k} P(y^{(i)} \mid x)}{\sum_{j=1}^{K} P(C_j)},$$

where $P(y^{(i)} \mid x)$ is the sequence-level probability of output $y^{(i)}$, computed as the average token probability across the sequence. For binary questions ($K = 2$), we report $\hat{p}_{\text{Yes}}$ as the final confidence estimate.

In practice, this approach does not yield benefits in EVOLVECAST, since binary Yes/No questions naturally collapse to two clusters. Thus, while clustering may offer richer signals in open-ended generation tasks (e.g., free-form QA or summarization), in binary forecasting it effectively reduces to re-labeling outputs without adding new information. Full quantitative results are shown in Table 4.

To further illustrate, Figs. 7 plots the confusion matrices for DS R1 Qwen-1.5B under both the baseline logits method and the clustering variant. The matrices show that clustering does not meaningfully shift the distribution of errors: the model still predicts "Still" excessively, and when it does move, the confusion between "Up" and "Down" persists.

## G.2 HUMAN FORECAST REFERENCE AS CONTEXT

We also test an ablation where the model is provided with the contemporaneous aggregate human forecast as an additional context feature. Prompts are augmented with a line such as: *"Human forecast for this question on $t$ is $h_t\%$."* This setting gives the model an explicit calibration anchor that, in principle, should simplify the task by showing where expert forecasters stood at the time.

The motivation is twofold. First, it simulates a collaborative human–AI forecasting scenario, where models can build on expert input rather than starting entirely from scratch. Second, it allows us to probe whether models meaningfully reason about how new evidence shifts beliefs relative to the anchor, or whether they simply mirror the human input.

Table 5 reports results under this setting for three DS R1 models using verbalized confidence (black-box) and direct directional prompting. While direct prompting again achieves higher directional agreement than verbalized probabilities, providing the human forecast reference itself does not yield any measurable improvement. This is somewhat surprising, as human forecasters rely heavily on such reference anchors; the lack of benefit here suggests that current models are unable to incorporate even strong external signals into their belief updating in a meaningful way.

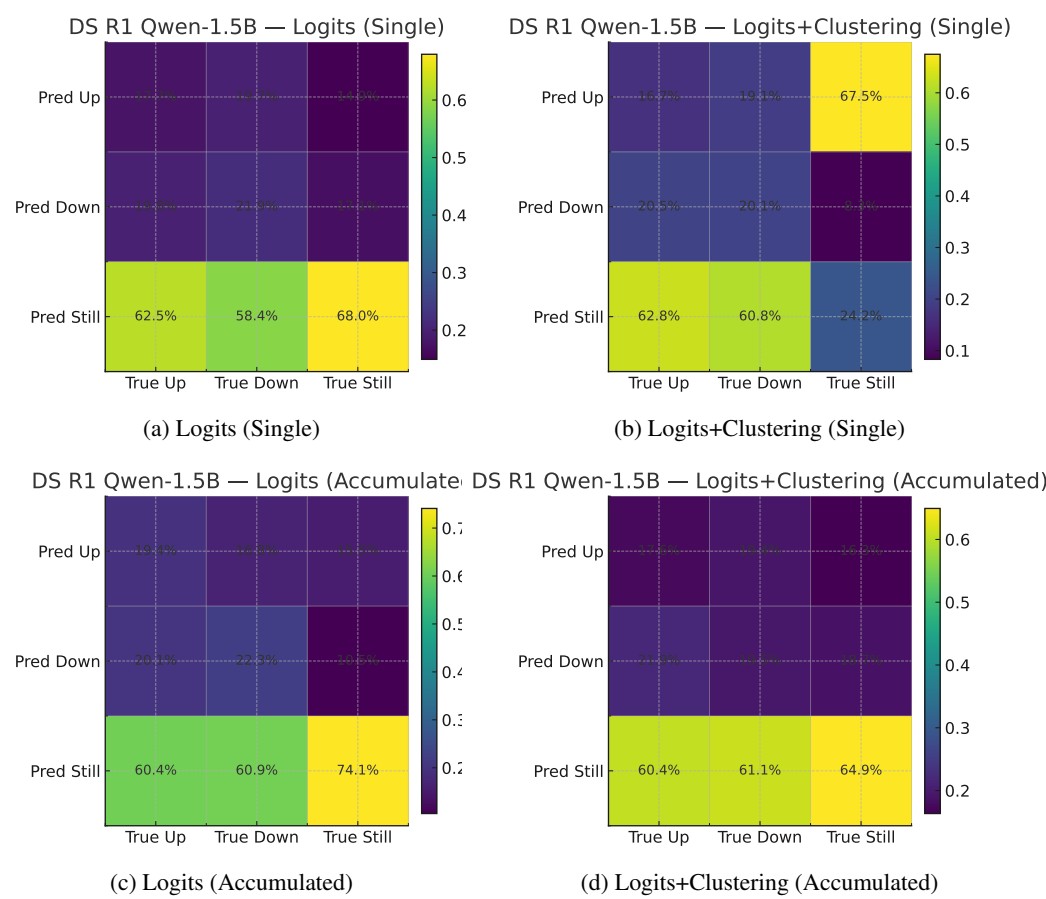

Figure 7: Normalized confusion matrices for DS R1 Qwen-1.5B under logits-based confidence and logits+clustering, for Single and Accumulated news. Values are column-normalized, showing $\Pr(\text{pred} \mid \text{true})$ (%). Clustering yields no qualitative change in error distribution.

Table 5: Results for **human forecast reference as context**, comparing verbalized confidence and direct directional prompting under the Single Update setting. Direct prompting consistently outperforms verbalized confidence, but including human forecasts as anchors produces no measurable gains.

| Model | Method | MDA | Precision | Recall | F1 |
|---|---|---|---|---|---|
| DS R1 Qwen-1.5B | Verbalized | 0.279 | 0.487 | 0.279 | 0.284 |
| | Directional QA | 0.350 | 0.441 | 0.350 | 0.376 |
| DS R1 Qwen-7B | Verbalized | 0.308 | 0.461 | 0.308 | 0.329 |
| | Directional QA | 0.457 | 0.462 | 0.457 | 0.457 |
| DS R1 LLaMA-8B | Verbalized | 0.307 | 0.479 | 0.307 | 0.327 |
| | Directional QA | 0.487 | 0.455 | 0.487 | 0.467 |

