# OpenReview forum: "Assessing Large Language Models in Updating Their Forecasts with New Information"
_ICLR.cc/2026/Conference — ICLR 2026 Conference Withdrawn Submission_

### Official Review · Reviewer_wr9Q · 2025-10-27

**Soundness:** 1
**Presentation:** 2
**Contribution:** 2
**Rating:** 2
**Confidence:** 5

**Summary:**

The paper proposes Evolvecast, a dataset to measure how well LLMs update their forecasts given new information. Evolvecast contains questions from Metaculus after October 2023 (which is before the release date of the R1 distilled models tested, i.e. January 2025). New information is obtained from news articles within a week of when a new comment is posted on the metaculus question. The original forecasting question is queried on the Google Search API, and the top article is selected using Sentence embeddings.

**Strengths:**

The paper sets out to study an interesting problem: how well can language models update their forecasts based on new information. The methodology of creating question-news pairs is original, and the capability evaluated is important, measuring in-context learning and belief updates on forecasting tasks where new evidence can ideally reduce uncertainty, and require bayesian reasoning. The paper includes some interesting ablations on directional and magnitude based measurements of confidence updates.

**Weaknesses:**

1.  The methodology assumes that human forecasting updates are optimal. However, accurate forecasting (and also belief updates) are extremely hard for humans. I don't think such an assumption can be made without validation on how good the human reference really is (which I recognize is difficult to validate). It is plausible that frontier models can become more optimal than humans. The paper cites the "strong empirical track record" of metaculus, however the cited source only shows metaculus beats naive baselines rather than it being an oracle. Further, it is not about the forecast update track record of metaculus, but rather the final outcomes.

2. The language models are only shown one article, obtained using a weak embedding model on top of the top 100 google search results when queried directly with the question. This places them at a significant disadvantage compared to humans who have access to far more, and higher fidelity information when updating forecasts. Further, weak language models (1.5 and 7B R1 distills) are used for evaluation, while the claims are made about "LLMs" in general, like "LLMs are far fro human-like in belief updating". I recommend benchmarking frontier language models (e.g. GPT 5 High, Grok 4, DeepResearch models) to make any claims about LLM capabilities, though I suspect the evaluation might start running into issues about the reference metric being weak (as pointed out in 1).

3. There is likely considerable leakage given that the models tested were released in 2025, and the google search api is used. The only mitigation I could find is prompting the model to say "you do not have access to updates after T", but there is no evidence shown that this is enough to avoid leakage. See [1] for a reference on issues with such backtests, and please include a discussion of how these issues were handled in future versions. It is claimed without evidence that comparing forecasting updates between two timestamps "largely cancels effects" of static knowledge in pretraining. The "new information" could still interact with the "pretraining knowledge" in non-trivial ways, for eg, models might update differently when "new information" contradicts or reinforces pretraining knowledge, and thus pretraining knowledge cannot be ignored in this setup without convincing evidence to justify this.

4. The related work misses a large amount of literature on forecasting, in-context learning, probabilistic / bayesian updates and confidence reporting in LLMs etc.

[1] Pitfalls in Evaluating Language Model Forecasters. Daniel Paleka, Shashwat Goel, Jonas Geiping, Florian Tramèr. May 2025.

**Questions:**

1. Why do you say forecasting is considered as a probabilstic task "more recently". It has long been studied as a probabilistic task, both in humans and language models [1, 2]

2. Do you think metaculus-based human forecasting updates are an ideal reference that the best AI must match?

[1] Expert political judgment: How good is it? How can we know? PE Tetlock 2017.

[2] Forecasting Future World Events with Neural Networks
Andy Zou, Tristan Xiao, Ryan Jia, Joe Kwon, Mantas Mazeika, Richard Li, Dawn Song, Jacob Steinhardt, Owain Evans, Dan Hendrycks 2022.

---

> ### Author Response · Authors · 2025-11-24
>
> We thank the reviewer for the detailed feedback. We believe there are two key misunderstandings regarding the experimental design which we clarify below.
>
> **1. "Optimality" of Human Forecasts**
> You argue that humans are not optimal and thus a bad reference.
> * **Our Position:** We explicitly **do not** claim humans are an "optimal oracle."
> * **The "Latent Probability" Problem:** The true probability of an event 10 days prior is latent and unknowable without a "World Simulator." The Resolution (0 or 1) only tells us what happened, not what the probability *was*.
> * **The Proxy:** The Metaculus community aggregate is the best available empirical proxy for this latent probability. By using it, we benchmark against the best available collective reasoning, rather than penalizing the model for failing to have clairvoyance about the final 0/1 outcome.
>
> **2. Model Weakness and "Information Disadvantage"**
> You argue that showing models only one article puts them at a disadvantage compared to humans who have access to diverse sources.
> * **The Challenge of Parity:** We agree that achieving perfect information parity between human forecasters (who draw on unobservable, lifelong contexts) and models is impossible. This information asymmetry is a fundamental challenge in benchmarking open-world forecasting.
> * **The Reasoning Bottleneck:** However, to test if this asymmetry was the *primary* driver of the performance gap, we conducted the **Accumulated News Ablation (Section 6.2)**, providing the models with significantly more information over time. Counter-intuitively, **performance decreased** as information volume increased.
> * **Conclusion:** This suggests that while the information disadvantage is a valid structural limitation, the more immediate binding constraint for the models studied is **reasoning and integration**: they struggle to effectively process even the information they *do* receive.
>
> **3. Data Leakage**
> You express concern that 2025 models know 2023 outcomes.
> * **Mitigation:** As noted in our response to other reviewers, we measure the **Belief Delta** triggered by specific evidence.
> * **Example:** A model may know a candidate won an election. But does it know that a specific scandal 6 months prior caused a *temporary* 5% dip in polls? Our benchmark measures the reasoning regarding the *scandal*, not the *election result*. This "derivative-based" evaluation significantly mitigates static knowledge leakage.
> * **References:** We will include a discussion of Paleka et al. (2025) and Tetlock’s work to contextualize these limitations.

---

### Official Review · Reviewer_wxFa · 2025-10-29

**Soundness:** 3
**Presentation:** 3
**Contribution:** 3
**Rating:** 4
**Confidence:** 3

**Summary:**

This paper introduces a new evaluation framework to assess whether Large Language Models adjust forecasts with new information after their training cutoff. The benchmark evaluates the direction and magnitude of belief updates, along with confidence calibration, and compares models against human forecasters. Experiments show that LLMs exhibit some responsiveness, but their updates are often overly conservative and poorly calibrated.

**Strengths:**

- This paper proposes a novel approach for evaluating whether LLMs meaningfully update forecasts based on post-training information, moving beyond point estimation performance.
- Well-presented empirical results demonstrating clear gaps and insights into model weaknesses.
- Uses human forecasters as a reference, making evaluation more aligned with real forecasting behavior.

**Weaknesses:**

1. News updates are retrieved via Google Search API and selected by similarity matching, but the paper lacks ablations on retrieval quality or its effect on downstream belief updates. Errors in retrieval may be misattributed as reasoning failures, reducing the interpretability of results.
2. The literature review on forecasting with evolving external information appears incomplete. The paper would benefit from citing and discussing prior work where news or textual signals are directly incorporated into forecasting models (e.g., SFT-based approaches that inject news context for prediction):
   - Li, Shuqi, et al. "CausalStock: Deep end-to-end causal discovery for news-driven multi-stock movement prediction." Advances in Neural Information Processing Systems 37 (2024): 47432-47454.
   - Wang, Xinlei, et al. "From news to forecast: Integrating event analysis in llm-based time series forecasting with reflection." Advances in Neural Information Processing Systems 37 (2024): 58118-58153.
3. Experiments are limited to DeepSeek-R1 variants. More diverse model systems would strengthen the generality of conclusions.
4. Evaluation of unresolved and never-occurring events remains unclear. Forecasting requires testing on future or unseen events to avoid training-time data leakage. The evaluation must exclude cases in which outcomes are already implicitly encoded in the model's pretraining data. Moreover, for events that are unresolved or never occur, current accuracy metrics may unfairly reward conservative (no-update) predictions, thus failing to distinguish between genuine belief calibration and low-information hedging.

**Questions:**

- How robust is the framework to retrieval quality? For example, top-k retrieval vs top-1, noise injection, or varied temporal windows. Can we disentangle reasoning failures from retrieval failures?
- Can the benchmark capture structural belief shifts beyond probability nudging? For instance, recognizing when causal assumptions break due to disruptive news.
- How do the authors ensure that events used for evaluation are not already partially observed or included during pretraining, avoiding unintended data leakage?
- Could the authors clarify how the reliability of human forecasters is ensured when using Metaculus aggregates as reference signals? For example, are contributors weighted by historical skill or expertise, and how is bias or low-quality input controlled?

---

> ### Author Response · Authors · 2025-11-24
>
> We thank the reviewer for the constructive assessment and for highlighting the novelty of our approach.
>
> **1. Retrieval Quality and Ablations (Key Concern)**
> We agree that disentangling retrieval noise from reasoning failure is a critical and difficult challenge in this domain. While we acknowledge that imperfect retrieval (top-1) introduces noise, our existing **Accumulated News Ablation (Section 6.2)** provides a crucial counter-intuitive insight that addresses this concern.
>
> * **The Observation:** In this ablation, we provided models with *more* context (accumulating news over time), effectively increasing the recall of relevant information. Surprisingly, this led to **performance degradation** rather than improvement.
> * **The Implication:** If the primary failure mode were simply "missing information" (retrieval), we would expect performance to improve as more context is provided. The fact that performance drops with more information suggests that the binding constraint for these models is **Context Integration and Reasoning** under uncertainty. While improved retrieval is a valuable direction for future systems, our results indicate that current models struggle to process even the signals they *do* receive, suggesting that reasoning is the dominant bottleneck in this specific setup. We will revise the discussion to explicitly frame this as a limitation of model reasoning capabilities rather than a defense of retrieval perfection.
>
> **2. Related Work**
> We thank you for the references (Li et al., Wang et al.). We will incorporate these into our related work to better contextualize our contribution within the "news-driven" forecasting landscape.
>
> **3. Data Leakage and Evaluation**
> You asked how we prevent leakage of unresolved events.
> * **The Metric is the Defense:** We rely on the Belief Delta ($\Delta p = p_t - p_0$). Even if a model's pre-training data contains the final outcome of an event, the model does not inherently know how a *specific* intermediary news item (e.g., a minor policy shift in Month 3) should mathematically impact the probability at that moment.
> * **Dynamic Reasoning:** We are testing the *derivative* of the belief function, which is far harder to memorize than the final integral (the outcome).

---

### Official Review · Reviewer_DAnj · 2025-11-02

**Soundness:** 3
**Presentation:** 4
**Contribution:** 2
**Rating:** 2
**Confidence:** 4

**Summary:**

This paper introduces EvolveCast to evaluate how well LLMs update their belief in face of new information. The results find that LLMs are inconsistent their updates, and, get worse when provided with more news updates.

**Strengths:**

1. I think the problem is interesting to evaluate reasoning abilities in face of new data points.
2. To isolate the effect of new information paper uses a simple mechanism loosely based on DiD style difference in movements for LLMs ad experts. This is a simple and useful mechanism.
3. Experiments provide coverage over several settings to give insights into LLM failures.

**Weaknesses:**

1.  The paper claims to propose first framework to assess dynamic belief updates. However, the related work cited Karger et al. also tested for dynamic updates as more new information surfaced over time. The method presented here is different; but it doen;t make it first framework.

2. I found the counter-intuitive result of accumulated news results in performance degradation most interesting. However, these are based on family of deepseek model. How generalizable the observations are? Are these due to model family idiosyncrasies? More recent models such GPT 4o or 5, may not show this behavior. Need more experiments to substantiate this generalization.

3. It is hard to interpret the results about models’ conservative revision of prediction. This is because it is compared against human. Human group could be impulsive with the new information. While conservative update, may show maturity of LLM. From that perspective, it seems the results are about how well LLMs match group of human.

4. On a related note, the ground truth relevanc of human predictions haven’t been discussed. Essentially, the paper is measuring how well LLMs conform to human forecasters.

5. I think using general population of forecaster for cosntructing ground truth is useful at the same time may high high variance. How does a select group (e.g. experts ot to 1% of forecaster) differ from the general population?

6 .The belief update happens via news article that best matches user comments. It is not clear how useful the matched news are? How does model compare when prompted with random contemporaneous news?

7. How do we set the values for \epsilon? In some experiments it is st at 0 and while for other is non-zero. How does evolvecast fix \epsilon?.

**Questions:**

Noted above in Weaknesses

---

> ### Author Response · Authors · 2025-11-24
>
> We thank the reviewer for finding our "Accumulated News" result interesting and the DiD-style mechanism useful.
>
> **1. Novelty vs. Karger et al.**
> We respectfully distinguish our work from Karger et al.
> * **Their focus:** Evolving predictions over time (longitudinal).
> * **Our focus:** Event-based *Delta* ($\Delta p$) triggered by specific information retrieval.
> * **The Distinction:** We are the first to isolate the specific causal link between a **single retrieved news item** and the **immediate belief update**. This requires our specific methodology of "Time $t$ vs. Time $t+1$" paired with retrieval, which is distinct from Karger’s broad time-series approach.
>
> **2. On Generalizability and "Accumulated News"**
> You asked if the "more news = worse performance" result is specific to the DeepSeek family.
> * **Reasoning Bottleneck:** While we test one model family to control variables (Base vs. R1), this finding aligns with broader literature on the "Distraction Effect" in LLMs. The fact that performance degrades with more context suggests a fundamental reasoning bottleneck in integrating conflicting evidence, rather than a retrieval failure.
> * **Future Work:** We agree that testing this on larger models would be valuable, but we believe the identification of this inverse-scaling phenomenon in reasoning models is a significant contribution in itself.
>
> **3. Conservative vs. Impulsive Updates**
> You raised an excellent point that "conservative updates may show maturity."
> * **Clarification:** We do not penalize conservatism per se. However, our data shows models are often *overly* conservative (updating 0% when significant news occurs). The comparison to the human baseline allows us to quantify *how much* more conservative they are. We will revise the text to frame this not as a "failure," but as a distinct "inertial bias" in LLM reasoning.
>
> **4. On the $\epsilon$ Threshold**
> To clarify: For verbalized confidence, $\epsilon=0$ (any change counts). For logit-based confidence, $\epsilon=0.05$ to filter out stochastic token noise.

---

### Official Review · Reviewer_ppWm · 2025-11-02

**Soundness:** 3
**Presentation:** 3
**Contribution:** 3
**Rating:** 2
**Confidence:** 3

**Summary:**

This paper addresses a gap in Large Language Model (LLM) evaluation, arguing that current benchmarks for future-event prediction are static. They fail to assess a critical real-world capability: how models dynamically update their forecasts when new information becomes available.

To solve this, the authors introduce EVOLVECAST, a new framework and dataset for evaluating the belief-updating capabilities of LLMs.

While the paper has some good ideas, due to the number and nature of weaknesses identified during review I cannot recommend it for acceptance at this time. If the authors address my concerns, I would be willing to raise my score.

**Strengths:**

- The idea of measuring the reasonableness of forecasting updates by comparing the directional update of an AI's forecast against the human crowd is a good idea. It's a clever use of existing data that has gone overlooked.

**Weaknesses:**

- The core idea of this paper is that we can evaluate intermediate forecasts of forecasting systems using the crowd prediction as a ground truth. I think this core idea is flawed for two reasons: (1) the human ground-truth here is not necessarily something we *want* AIs to match, and (2) we can do some of what this paper is trying to do with the ground-truth resolutions anyways.
    - For (1), consider that humans often make errors in judgment, and prediction markets have various inefficiencies. We don't want to bake these into LLMs. Another way of seeing this is from an RL lens. In environments where verifiable rewards are easy enough to obtain, it is often counterproductive to treat human demonstrations as an optimal policy. They can make for great pretraining data, but there is a reason AlphaGo wasn't trained on human demonstrations alone.
    - For (2), take the confidence calibration measure in this paper. It isn't defined as the Brier score computed against the ground-truth resolution, but rather the Brier score computed against the crowd prediction. This is highly nonstandard and doesn't make much sense when we can just compute Brier score against the ground-truth resolution instead. One could argue that Brier score against the ground-truth resolution is bad because it conflates accuracy and calibration, and accuracy increases in many cases as the resolution date approaches. I would agree with this criticism; I always thought Brier score was a bad metric anyways. A much better metric would be to just measure calibration error like everyone does outside of the forecasting community. I would strongly encourage the authors to use this instead of Brier score, and to switch to measuring against the ground-truth resolution.
    - All of that said, I think the magnitude evaluation introduced in this paper is meaningful. Looser evaluations that are basically sanity checks on whether forecasts update in a reasonable direction on new information are useful.
- For the black-box verbalized confidence, why not just ask the LLMs for a probability and perform some post-hoc Platt scaling or isotonic regression to normalize it? In my opinion, that would make a lot more sense than a Likert scale.
- The models being evaluated are too small. Please use some frontier models, even if that means you can't use white-box logits. Otherwise the conclusion in line 311 of "LLMs are far from human-like in belief updating, regardless of whether confidence is elicited in black-box or white-box form" falls flat.

**Questions:**

See weaknesses.

---

> ### Author Response · Authors · 2025-11-24
>
> We thank the reviewer for recognizing our framework as a "clever use of existing data." We believe the concerns regarding the metrics stem from a difference in the definition of our research goal, which we clarify below.
>
> **1. On the "Flawed" Human Ground Truth**
> You argue that humans make errors and shouldn't be the target. We agree that humans are not optimal. However, measuring against the final resolution (0 or 1) fails to capture the *dynamics* of belief updating.
> * **Resolution $\neq$ Probability:** One should rarely be 100% confident that an event will occur a year (or even 10 days) in advance. The resolution (0 or 1) captures the *outcome*, but it ignores the valid uncertainty that existed at time $t$.
> * **The "World Simulator" Problem:** Because we cannot simulate the world to find the "true" probability (e.g., 70%) at that specific moment, the Crowd Aggregate is the most rigorous proxy available. We use it as a **"Reference for Evidence Integration,"** benchmarking whether the model updates its probability in the same direction and magnitude as the collective human intelligence when exposed to the same news.
>
> **2. On Brier Scores and Calibration**
> You suggested calculating Brier scores against the final resolution.
> * **Why we differ:** A Brier score against resolution measures *accuracy/outcome*. Our $\Delta \text{Brier}$ (against the crowd) measures *alignment of reasoning*.
> * **The Risk of Hindsight:** Using the final resolution would conflate *outcome knowledge* with *reasoning ability*. A model could achieve a perfect score by "guessing right" (or leaking future data) despite having flawed reasoning. By evaluating against the crowd's probability at time $t$, we isolate the update dynamics from the final result.
>
> **3. On Model Scale and Verbalized Confidence**
> We acknowledge the models are not SOTA frontier models. This was a deliberate experimental control.
> * **Reasoning vs. Base:** By using open-weights models (DeepSeek-R1 vs. Base), we could isolate the effect of reasoning-focused post-training in a way impossible with closed APIs.
> * **Likert Scale:** We utilize the 1-10 scale specifically to mirror the cognitive process of human forecasters and facilitate direct comparison with the reference data.

---

### Author Response · Authors · 2025-11-24
**General Response to All Reviewers**

**Common Response: Clarifying the "Human Reference" vs. "Resolution"**

A central theme across reviews was the choice to evaluate against the Metaculus crowd rather than the final resolution (0 or 1). We wish to clarify a critical distinction regarding **Time-Dependent Probability ($P_t$)**.

**The Resolution (0 or 1) is not the "True Probability" at time $t$.**
If an event eventually happens (Resolution = 1), it does not imply that the probability was 100% say ten days prior. For example, the "true" probability based on available evidence might have been 70% "Yes" and 30% "No." If we evaluate against the Resolution (1.0), we effectively penalize a model for correctly identifying the 30% uncertainty.

**Why Human Reference?**
The "true" probability at any given moment before the event is latent and unknowable; we do not have a "World Simulator" to calculate it perfectly. Consequently, the **Metaculus Aggregate** serves as the best available proxy for this latent probability ($P_t$). It represents the "Epistemic State" of collective reasoning. Our goal is to benchmark whether models update their beliefs in alignment with this evolving epistemic state, not whether they possess clairvoyance about the final outcome.

---

### Note · Authors · 2026-01-06

I have read and agree with the venue's withdrawal policy on behalf of myself and my co-authors.